# The ABA/LANCL Hormone/Receptor System in the Control of Glycemia, of Cardiomyocyte Energy Metabolism, and in Neuroprotection: A New Ally in the Treatment of Diabetes Mellitus?

**DOI:** 10.3390/ijms24021199

**Published:** 2023-01-07

**Authors:** Sonia Spinelli, Mirko Magnone, Lucrezia Guida, Laura Sturla, Elena Zocchi

**Affiliations:** Section of Biochemistry, Department of Experimental Medicine, University of Genova, Viale Benedetto XV 1, 16132 Genova, Italy

**Keywords:** AMPK, PGC-1α, Sirt1/3, LANCL1/2, mitochondrial proton gradient, hypoxia, oxidative stress, neurodegeneration, NO, cardioprotection

## Abstract

Abscisic acid (ABA), long known as a plant stress hormone, is present and functionally active in organisms other than those pertaining to the land plant kingdom, including cyanobacteria, fungi, algae, protozoan parasites, lower Metazoa, and mammals. The ancient, cross-kingdom role of this stress hormone allows ABA and its signaling pathway to control cell responses to environmental stimuli in diverse organisms such as marine sponges, higher plants, and humans. Recent advances in our knowledge about the physiological role of ABA and of its mammalian receptors in the control of energy metabolism and mitochondrial function in myocytes, adipocytes, and neuronal cells allow us to foresee therapeutic applications for ABA in the fields of pre-diabetes, diabetes, and cardio- and neuro-protection. Vegetal extracts titrated in their ABA content have shown both efficacy and tolerability in preliminary clinical studies. As the prevalence of glucose intolerance, diabetes, and cardiovascular and neurodegenerative diseases is steadily increasing in both industrialized and rapidly developing countries, new and cost-efficient therapeutics to combat these ailments are much needed to ensure disease-free aging for the current and future working generations.

## 1. The ABA/LANCL1/2 Hormone/Receptor System

### 1.1. Abscisic Acid, an Ancient Stress Signal

Abscisic acid is an ancient signal molecule, which likely evolved in primordial unicellular organisms, much earlier than the divergence of plants and animals. It is present and active in modern bacteria, unicellular algae, basal and vascular plants, as well as mammals, including humans. In the more complex Metaphyta and Metazoa, ABA has a conserved role as a stress hormone, adapting cell, tissue, and organism functions to environmental stimuli; it is perhaps the oldest cross-kingdom hormone known to date.

Although the molecule itself is simple compared with other hormones (an isoprenoid structure with a mol. weight of 264, Figure 1), the perception and signaling pathways of ABA are quite complex and distinct in divergent organisms, such as plants and mammals. At the functional level, however, strikingly conserved effects of ABA can be observed in organisms as evolutionarily distant as land plants and mammals.

For instance, ABA leads to nitric oxide (NO)-mediated stomatal closure in response to several stressors, including UV light, in plants [2], and human keratinocytes exposed to UV light release ABA, which stimulates NO production by these cells [3]. In human innate immune cells, physical or chemical stimuli induce ABA release and ABA stimulates NO production among other functional responses, such as migration and phagocytosis [3]. It could be argued that NO may be another of those few molecular signals that evolved very early in primordial cells. This ABA/NO-mediated “alarm pathway” in response to an environmental stress is indeed conserved across kingdoms [4].

The quest for plant and animal ABA receptors has been underway for decades and is still a topic of intense investigation; recent evolutionary studies indicate that ABA was present before its (modern) specific receptor PYL acquired the ability to be hormonally modulated by it, as occurs in land plants [5].

### 1.2. Mammalian LANCL Proteins

LANCL (Lan C-like) proteins derive their name from their sequence homology with bacterial lanthionine synthetase, which synthesizes lantibiotics, a class of natural antibiotics originating from modified cysteine. The mammalian LANCL family comprises three proteins: LANCL1 is the most highly expressed, particularly in the brain, and is a cytosolic protein also loosely associated with the plasmamembrane [6]. LANCL2 is membrane-anchored through its myristoylated N-terminal [7], and LANCL3 has the lowest expression levels of the LANCL proteins and may be a pseudogene. The three LANCL proteins share a significant (approx. 35%) sequence identity, likely arising from gene duplication. As the triple knockdown in mice does not reduce the brain content of a downstream metabolite of lanthionine, lanthionine ketimine, it has been concluded that mammalian LANCL proteins are not involved in lanthionine synthesis [8]. Interestingly, however, triple LANCL knock-out (KO) mice die prematurely [9], indicating an important role for LANCL proteins in animal physiology, which may be linked to the features described in this review.

LANCL proteins also show a significant (approx. 30%) sequence identity with another evolutionary distant protein, the plant orphan, G-protein linked receptor GCR2, which was proposed as an ABA receptor in Arabidopsis [10], a discovery subsequently obscured, but not dismissed [11,12], by the discovery of the PYR/PYL/RCAR family of ABA-sensing proteins in higher plants [5].

Still, the sequence homology between plant GPCR-type ABA receptor and mammalian LANCL proteins spurred research on their possible role as mammalian ABA receptors.

#### 1.2.1. The Membrane-Bound LANCL2 Receptor

The LANCL2 protein first attracted interest as a putative mammalian ABA receptor; indeed, in vitro studies using several different techniques demonstrate high-affinity ABA binding to human recombinant LANCL2 (Kd 2.6 nM) [13], and LANCL2 is required for ABA action in several mammalian cell types [14]. LANCL2 has an unusual behavior as a hormone receptor: it is coupled to a G protein when membrane bound, but it is de-myristoylated upon ABA binding, detaching from the membrane and accumulating in the cell nucleus [15]. This behavior combines features typical of the receptors for peptide (G protein coupling) and for steroid hormones (nuclear translocation), perhaps a heritage of the primordial origin of the hormone, when Nature was still experimenting with its signaling pathways, or the result of the hydrophilic or lipophilic nature of dissociated or of protonated ABA, respectively. The binding of ABA to LANCL2 on the inner plasma membrane layer requires influx of the hormone through the plasma membrane, which occurs through transporters of the anion exchanger (AE) superfamily [16]. Protonated ABA can instead diffuse through the lipid bilayer; however, a very low percentage of ABA is protonated at the near-neutral pH of plasma and interstitial fluids. Thus, the exchange of ABA between extra- and intracellular fluids requires a transport system. An area where ABA transport is not needed to cross barriers is the gastric environment, where the strongly acidic pH favors the protonated, membrane-permeant form of the molecule, allowing diffusion of ABA across the gastric lipid bilayer. This fact probably accounts for the rapid absorption of the hormone after oral intake [17].

#### 1.2.2. LANCL1 Is Also an ABA Receptor

A reduction, though not the complete abrogation, of the effect of ABA occurs in adipose and muscle cells after LANCL2 silencing [18,19], suggesting a role for other receptors in the metabolic action of the hormone. A more direct indication that other receptors could contribute to mediate the metabolic actions of ABA comes from studies on LANCL2 KO mice. LANCL2 KO mice show a reduced glucose tolerance compared with wild-type siblings; however, they are still responsive to exogenous ABA (1 μg/kg body weight (BW)/day), which significantly reduces the AUC of glycemia after glucose load to values similar to those of wild-type animals [19]. On the one hand, these results indicate that the absence of LANCL2 impairs glucose tolerance; on the other hand, they indicate that another receptor can substitute for LANCL2 in the stimulation of muscle and adipose tissue glucose transport, although at higher ABA concentrations than those reached by the endogenous hormone. Indeed, upon intake of ABA at a dose of 1 μg/kg BW plasma, ABA (ABAp) increases between 10 and 50 times compared with endogenous levels in humans [17]. The increase in ABAp achieved by exogenous ABA supplementation can apparently recruit a receptor with lower affinity, which may not be activated by lower endogenous ABA levels. An obvious candidate for this role is LANCL1, as it shares a significant sequence identity (54.2%), similar intracellular localization (it is membrane-associated, although not membrane-bound), and tissue expression pattern with LANCL2.

Interestingly, silencing or genetic ablation of LANCL2 in cells, or in mice, results in the spontaneous overexpression of LANCL1. Similarly, silencing of LANCL1 results in a significant increase in the expression of LANCL2 in L6 muscle cells [19]. These observations, together with the redundancy of ABA receptors, point to the physiological relevance of the ABA/LANCL hormone/receptor system in mammals. From a functional perspective, LANCL1 binds ABA with a somewhat lower affinity compared with LANCL2, but it activates the same signaling pathway (the AMPK/PGC-1α/Sirt1 axis), resulting in similar transcriptional and functional responses (increased glucose uptake and metabolism, mitochondrial respiration, and uncoupling), in vitro in rat myoblasts and in vivo in the skeletal muscle of LANCL2 KO mice [19].

#### 1.2.3. AMPK Activation Downstream of LANCL1/2

Experiments performed in vitro on L6 myoblasts and ex vivo on murine skeletal muscle demonstrate that ABA stimulates muscle glucose transport in the absence of insulin and that activation of AMPK is responsible for this effect, as it is abrogated by the inhibition of AMPK with dorsomorphin. ABA indeed increases the phosphorylation of AMPK on Thr172 in L6 cells and in mouse muscle and also stimulates AMPK transcription [20]. Preincubation of L6 myoblasts with AZD5363, at pan-Akt competitive inhibitor, significantly increased pAMPK levels in ABA-treated compared with untreated cells, indicating the presence of an inhibitory effect by Akt on ABA-induced AMPK activation [20]. Indeed, the activation of Akt by phosphorylation on both Thr308 and Ser473 inhibits AMPK phosphorylation on Thr172 by LKB1 [21,22]. Akt lies at the crossroads between the starved and fed state: in the fed state, insulin favors double phosphorylation (on Ser473 and on Thr308), and maximal activation of Akt [22], which triggers cell energy storage, mainly via glycogen and triglyceride synthesis, concomitantly inhibiting AMPK phosphorylation and activation by LKB1. Conversely, AMPK activation occurs under conditions of reduced cell energy availability and stimulates metabolic and mitochondrial functions aimed at restoring energy balance. Stimulation of cell glucose uptake is a common feature of both ABA- (via AMPK) and insulin- (via Akt) triggered signaling, enabling energy production and energy storage, respectively. Glucose uptake, detected in skeletal muscle by dynamic micro-PET, increases 2-fold in ABA-treated compared with untreated rats during an oral glucose load [20]. Given the high percentage of BW represented by skeletal muscle in rodents, approximately 45%, the increased muscle glucose uptake is likely responsible for the accelerated blood glucose clearance observed in ABA-treated compared with control mice.

#### 1.2.4. ABA Signaling in Skeletal Muscle

The signaling pathway activated by ABA in the skeletal muscle involves the AMPK/PGC-1α/Sirt1 axis, resulting in the increased gene transcription and protein overexpression of the glucose transporters GLUT1 and GLUT4, of the NAD-synthesizing enzyme Nampt, of RabGAP TBC1D1, and of the muscle-specific mitochondrial uncoupling proteins UCP-3 and sarcolipin and in an increased mitochondrial DNA content [19]. These transcriptional and translational effects of ABA increase muscle glucose uptake and energy metabolism, leading to increased muscle glucose consumption. The increased muscle expression of LANCL1 in LANCL2 KO mice as compared with wild-type siblings may explain why LANCL2 KO mice with streptozotocin (STZ)-induced diabetes indeed respond to ABA similarly to, or perhaps even better than, wild-type mice. In addition, LANCL2 KO mice have a higher skeletal muscle mitochondrial DNA content and increased expression levels of AMPK, PGC-1α, GLUT1/4, Nampt, and UCP-3 compared with wild-type mice, levels which further increase after chronic ABA administration. We observed a significant increase in the transcription of key glycolytic enzymes (GaPDH, PFK1) and of PDH in the skeletal muscle of ABA-treated wild-type and LANCL2 KO mice as compared with untreated controls, which is expected to stimulate oxidative muscle glucose consumption. Interestingly, LANCL1-overexpressing LANCL2 KO female mice fed a high-glucose diet for three months had a significantly lower body weight gain as compared with wild-type siblings under higher food intake. These findings suggest that muscle and adipocyte mitochondrial uncoupling and increased oxygen consumption controlled by the ABA/LANCL system may affect whole body energy consumption. At the end point of a similar protocol of ABA-pretreatment followed by diabetes induction with low-dose STZ, the mean glycemia of LANCL2 KO mice was significantly lower than that of wild-type animals. These effects can be attributed to the overexpression of LANCL1, which can substitute for LANCL2 in binding ABA and activating its signaling pathway.

Finally, the increased transcription of the insulin receptor mRNA in the skeletal muscle from chronically ABA-treated mice suggests that long-term treatment with ABA may improve muscle sensitivity to both endogenous and exogenous insulin. Probably as a consequence of the increased glucose uptake and oxidation, a markedly increased physical performance on a running wheel was observed in ABA-treated mice compared with untreated controls [20].

#### 1.2.5. The ABA/LANCL System in the Adipose Tissue

Adipose tissue (AT) is one of the largest organs in the body and plays an important role in energy balance and glucose and lipid homeostasis. In mammals, white adipose tissue (WAT), diffused subcutaneously and abdominally, is specialized in energy (triglyceride) storage; conversely, brown adipose tissue (BAT), which is much less abundant than WAT and has specific localization sites (cervical, supraclavicular, paravertebral, and supra-adrenal in adult humans), is specialized in energy expenditure. Excess visceral WAT is currently believed to be conducive to the metabolic syndrome, overt diabetes, and cardiovascular disease due to a state of low-grade inflammation originating in, and maintained by, adipokines produced in WAT and cytokines produced by WAT-infiltrating inflammatory cells. Thus, the observation that ABA can reduce WAT inflammation in mice fed a high-fat diet [23] is particularly important in view of the possible use of ABA-containing nutraceuticals to combat low-grade inflammation in the adipose tissue. Pointedly, high-sensitivity C-reactive protein levels (hs-CRP, a measure of chronic, low-grade inflammation) were significantly reduced in prediabetic subjects after treatment with an ABA-rich vegetal extract compared with untreated controls [24]. Thermogenesis by BAT (the so-called non-shivering thermogenesis) is activated by cold stress and is controlled by sympathetic innervation and by several hormonal signals (T3, adenosine): heat production occurs due to the partial uncoupling of mitochondrial oxidative phosphorylation during the oxidation of glucose and of triglyceride-derived fatty acids. These features of BAT make it an attractive target for therapeutic strategies aimed at stimulating BAT activity to reduce body weight. Alas, at variance with WAT, BAT only accounts for a minute percentage of total AT; thus, another currently pursued strategy to increase whole-body energy expenditure is to induce the development of brown fat-like cells (also known as beige cells) in WAT. Recent studies have demonstrated that an increase in beige adipocytes in WAT enhances whole-body energy expenditure and is expected to reduce the risk of diet-induced obesity and metabolic diseases [25].

#### 1.2.6. Mitochondrial Effects of the ABA/LANCL System

As mitochondria are the main energy producers in mammalian cells, it comes as no surprise that the ABA/LANCL system stimulates several mitochondrial functions, including mitochondrial biogenesis, oxidative metabolism, oxygen consumption, and proton gradient accumulation, in both adipose and muscle cells.

In murine and human preadipocytes, in vitro ABA does not induce triglyceride accumulation; instead, it increases the mitochondrial content, O_2_ consumption, and CO_2_ production sustained by LANCL2-dependent increased GLUT4 expression and glucose uptake and stimulates the transcription of “browning” genes, such as uncoupling protein-1 (UCP-1) [18]. In vivo, one-month-long ABA treatment at 1 µg/Kg BW/day significantly increased the mitochondrial DNA content in the WAT and in WAT-derived preadipocytes differentiated in vitro from treated mice compared with untreated controls. ABA also increased the expression of mitochondrial uncoupling proteins 1 and 3 (UCP-1/3) in brown adipose tissue from treated mice and stimulated BAT glucose uptake, an indirect measure of BAT thermogenesis [18].

In rodent L6 myocytes, LANCL1/2 overexpression and/or ABA treatment stimulated mitochondrial biogenesis, as witnessed by increased mitochondrial DNA and O_2_ consumption [19].

#### 1.2.7. Non-Overlapping Roles of ABA and Insulin in Energy Metabolism

Based on the above results, a non-overlapping role for ABA and insulin in muscle and adipocyte metabolism can be envisaged. Insulin induces the phosphorylation of Akt, inhibiting AMPK and shifting the metabolic program from the starved to the fed state, activating glucose transport and metabolism, and storing excess energy abundance via glycogen, protein, and lipid synthesis. ABA instead induces the phosphorylation of AMPK, thereby activating the metabolic response to starvation and/or low glucose availability. This response includes the stimulation of glucose transport and oxidation for energy production, mitochondrial biogenesis, respiration, and uncoupling (Figure 2).

Altogether, this background allows us to identify the ABA/LANCL system as a new player in the control of glycemia homeostasis and of mitochondrial energy production and a possible new therapeutic target in pathological conditions such as dysfunction, diabetes mellitus, and hypoxia. Indeed, hypoxia is inextricably linked to diabetes due to the micro- and macro-vascular damage induced by the metabolic derangement.

## 2. The ABA/LANCL System in Diabetes

### 2.1. Plasma ABA in Diabetic Subjects

In mammals, two peptide hormones are released by glucose/nutrient-sensing cells: pancreatic islet β-cells release insulin and intestinal endocrine cells release glucagon-like peptide 1 (GLP-1). Among other actions, GLP-1 contributes to stimulate insulin release and inhibits secretion by pancreatic α-cells of glucagon, the principal hormone activated by low blood glucose levels. ABAp also increases after an oral glucose load in healthy humans but not in subjects with type 2 diabetes (T2D) or with gestational diabetes (GDM) [26]. In GDM, resolution of the diabetic state that follows childbirth is accompanied by the restoration of normal ABAp, suggesting a critical role for ABAp in the maintenance of a normal glucose tolerance and a new possible ABA-centered pathogenetic mechanism that may underlie the diabetic condition. Indeed, the identification of a second hormone beside insulin capable of stimulating muscle glucose uptake would have significant consequences in diabetes mellitus, where insulin deficiency or insulin resistance reduce glucose tolerance. Together with insulin, ABAp is also undetectable or very low in type 1 diabetes (T1D) patients, suggesting that β-cells are the principal source of endogenous ABA in humans; thus, the demise of β-cells in T1D greatly reduces the availability of both hormones regulating glycemia, only one of which is currently replaced by therapy. As the metabolic actions of insulin and ABA are distinct, supplementation with one hormone cannot restore the functions of the other.

Indeed, low-dose oral ABA reduces glycemia and insulinemia in rats and in healthy humans undergoing a glucose load. The glycemia-lowering action of low-dose ABA in vivo reduces stimulation by hyperglycemia on β-cells and consequently insulin release. At variance with insulin, ABA does not induce hypoglycemia, even at a dose 100,000 times higher than the one efficacious in reducing glycemia (100 mg/Kg BW vs. 1 µg/Kg BW); thus, it has a very high therapeutic index; the absence of hypoglycemic risk due to excess dosage places ABA at significant variance with respect to insulin and to oral hypoglycemic drugs.

Intriguingly, the LANCL1 gene is located within the Insulin-dependent diabetes (Idd) 5.3 locus, which provides resistance to T1D in NOD mice [27]. LANCL1 is also among the candidate genes responsible for an observed complex phenotype of impaired neuronal function due to a microdeletion in the chromosomal region 2q34 [28].

ABA can be administered orally, it is readily absorbed because in the acidic gastric environment the protonated molecule is membrane permeant, and its plasma concentration remains high for several hours after intake [17], probably due to its binding to plasma proteins, which reduces renal clearance.

### 2.2. Clinical Studies on Borderline and Prediabetic Subjects

Indeed, clinical studies performed on healthy subjects with borderline values for metabolic syndrome [29] or prediabetes [24] indicate that low-dose ABA reduces glycemia, lipidemia, cardiovascular risk parameters, and low-grade inflammation after daily chronic administration.

In subjects with borderline values for metabolic syndrome (HbA1c and FPG values, TC, WC, and BMI) low-dose ABA supplementation (1 µg/Kg BW/day) for 75 days by means of a vegetal extract titrated in ABA reduced fasting glycemia and insulinemia, glycemia AUC after glucose load, HbA1c, TC, and body weight. As a consequence, the 10-year cardiovascular risk was significantly reduced [29]. In parallel with the human study, employing a vegetal extract as the source of ABA, mice fed a high-glucose diet were treated with the pure ABA molecule at the same daily dose as humans (1 µg/Kg BW/day) for four months, resulting in an improvement of glucose tolerance and a reduction of HbA1c, of blood lipids, and of body weight in the ABA-treated animals compared with untreated controls.

In prediabetic subjects (IFG or IGT) low-dose ABA supplementation improved glyco-metabolic and inflammation parameters [24]. ABA treatment did not significantly modify the anthropometric parameters, but a reduction of TC, LDL-C, and Tg was observed, indicating a similar downward trend as observed in borderline subjects. ABA supplementation also significantly reduced hs-CRP levels, demonstrating an improvement in the inflammatory status of prediabetic patients. Indeed, chronic low-grade inflammation lies at the heart of the pathogenetic mechanism underlying insulin resistance.

Based on the results of these studies, it is possible to conclude that the improvements in metabolic and bodily parameters observed with the food supplements were due to ABA in the vegetal extract of the compositions because similar results were observed in mice fed the pure molecule.

In another clinical set-up, a single dose of an ABA-containing vegetal extract was tested on the glycemia profile after intake of a standardized carbohydrate-rich breakfast. In each subject, breakfast with the food supplement significantly reduced the mean glycemia profile and the mean AUC of glycemia compared with the same breakfast without the supplement [29]. Ingestion of the ABA-rich extract increased ABAp 5- to 16-fold over fasting levels (5–15 nM), indicating that oral ABA is absorbed readily and contributes to the endogenous ABAp pool [17]. Reduction of post-prandial glycemia and insulinemia in normal subjects was also reported after intake of a different vegetal extract, titrated in ABA [30], indicating that the nature of the vegetal extract is irrelevant, as long as it contains a sufficient amount of ABA.

Together, these data outline an important role for the ABA/LANCL system in the physiology of glucose metabolism and energy metabolism in mammals. This background allows us to hypothesize a beneficial role for ABA in conditions were glucose metabolism and/or mitochondrial energy production are deranged. The extent to which insulin and ABA synergize to control glycemia is an open field of investigation. In both T1D and T2D, a severe reduction of endogenous plasma ABA or of the plasma ABA response to hyperglycemia occurs [26]. ABA appears to be mainly produced by β-cells, as plasma ABA is very low or undetectable in T1D subjects. Another possible explanation for this observation is that insulin promotes the release of ABA (also) from other cell types; in any case, a severe reduction or the complete destruction of β-cells could affect the release of both hormones uniquely endowed with the ability to stimulate muscle glucose uptake.

### 2.3. Oral ABA Ameliorates Glycemia in Insulin-Deficient Mice

The genetic ablation of Ca^2+^-permeable non-selective cation channel TRPM2 results in defective insulin secretion in CD1 mice; consequently, TRPM2 KO mice fed a high-glucose diet develop hyperglycemia due to insufficient insulin release [31]. In hypoinsulinemic TRPM2 KO mice, treatment with ABA, both at a single dose together with a glucose load, or chronically, in high-glucose-fed mice, reduced the glycemia profile and increased muscle glycogen storage without increasing plasma insulin levels [20].

Another murine model of insulin deficiency is the STZ-induced β-cell loss that mimics the demise of the β cell reservoir occurring due to autoimmune aggression in T1D. This experimental model showed a gradual loss of β cells over several weeks, or instead, a rapid induction of almost total β cell loss and consequent hyperglycemia over a short period of days depending on the dose and timing of STZ administration, i.e., multiple low-dose STZ (MLD) or single high-dose STZ (SHD). Recent studies have tested that treatment with ABA could improve glycemic control in these murine models of T1D, either alone, during the progressive β cell loss induced by MLD-STZ, or in addition to insulin, under conditions of complete endogenous insulin deficiency (SHD-STZ). The MLD and SHD-STZ protocols mimic the relative insulin deficiency observed in T2D and the absolute insulin deficiency of T1D, respectively.

In the MLD protocol of T1D induction, chronic ABA treatment improved the glycemic profile in treated mice compared with untreated controls during a 28-day period without a significant difference between plasma insulin levels after a final OGTT and with similar residual amounts of pancreatic insulin mRNA at the end-point. Thus, the improvement of glycemic control in the ABA-treated animals was not attributed to higher endogenous insulin levels but rather to the glycemia-lowering action of ABA via an increased skeletal muscle glucose uptake. Furthermore, ABA-treated mice had increased expression of the insulin receptor in skeletal muscle, suggesting an improved action of residual endogenous insulin [32].

In the SHD protocol, ABA alone could not substitute for insulin under conditions of total insulin deficiency when glycemia increased to higher than 500 mg/dL, but it improved the effect of exogenous insulin, when the dose of the peptide hormone was insufficient to restore euglycemia [32].

## 3. The ABA/LANCL System Protects Cardiomyocytes from Hypoxia

The brain and heart are eminently aerobic tissues, relying entirely on mitochondria for energy production.

When cardiomyocytes or neuronal cells experience insufficient O_2_ supply, reactive oxygen species (ROS) are overproduced, leading to the development of mitochondrial quality control disorders [33], defective oxidative phosphorylation, lipid peroxidation, and mitochondrial damage, eventually causing cell apoptosis. Interestingly, after the brain, the heart is the organ with the highest expression levels of LANCL1/2. In particular, LANCL1 expression in the heart is among the highest in non-neurological tissues and is approximately 4-times higher compared with LANCL2 expression [34]. Indeed, an important role for the LANCL proteins in cardiomyocyte mitochondrial function has recently been discovered.

### 3.1. The ABA/LANCL System Regulates NO Production in Rodent Cardiomyocytes

ABA is released from several mammalian inflammatory cell types after chemical or physical stress, suggesting that it might have a similar effect on cardiomyocytes. NO is particularly important in the heart as it profoundly affects myocardial function through electrical transmission, mechano/chemo-transduction, energy metabolism, and myocyte growth and survival. NO deficiency is associated with several heart diseases [35], and NO replacement therapy is being advocated as a means to improve cardiac performance [36,37]. Recent results indicate that the ABA/LANCL system indeed plays a hitherto unrecognized role in the regulation of NO production in cardiomyocytes.

#### 3.1.1. ABA Is Released by H9c2 under Hypoxia and Stimulates NO Production

The ultimate stressor for an eminently aerobic tissue such as the heart is hypoxia; indeed, the culture of rat H9c2 cardiomyocytes under hypoxic conditions results in ABA release and ABA in turn stimulates NO production [34]. ABA also stimulates cardiomyocyte glucose uptake and oxidation, increased O_2_ consumption under normoxia, and conservation of the mitochondrial proton gradient (ΔΨ) after hypoxia. Glucose oxidation provides ATP, but also NADPH, which increases in ABA-treated cells and is a pivotal actor in combating oxidative stress, via glutathione reductase, but it is also needed for NO production.

#### 3.1.2. ABA Stimulates eNOS Transcription, Expression, and Phosphorylation

In the heart, both Akt and AMPK can phosphorylate and activate eNOS, leading to the synthesis of NO, which improves mitochondrial function and protects cell viability. This apparent redundancy is likely related to the necessity by the heart to integrate different signals adapting its metabolism to changing conditions of nutrient availability and energy requirements. Akt, which lies downstream of insulin, promotes glucose uptake by the heart when blood glucose levels are high, but AMPK can also stimulate glucose uptake by phosphorylating the same targets as Akt in response to stress or hypoxia. Stimulation by ABA of eNOS in H9c2 cells indeed occurs via both AMPK- and Akt-dependent phosphorylation of eNOS. In addition to increasing eNOS transcription, protein expression and post-translational activation, ABA also increases the expression of the mitochondrial arginine transporter and of the regulatory enzyme for synthesis of tetrahydrobiopterine, a coenzyme required for the synthesis of NO from arginine, and reduces the expression of arginase, an enzyme competing with NOS for its substrate arginine [34].

#### 3.1.3. The ABA-Induced Increase in NO Improves Survival of H9c2 under Hypoxia

As a result of its stimulation of glucose uptake and metabolism and improved mitochondrial proton gradient restoration after hypoxia, ABA treatment significantly increases cardiomyocyte survival under hypoxia/reoxygenation in vitro.

#### 3.1.4. Transcriptional Control of LANCL1/2 on eNOS Transcription, Expression, and Function under Normoxia and Hypoxia

The overexpression of either LANCL1 or LANCL2 activates, while their combined silencing significantly reduces, eNOS transcription, protein expression, and activity in H9c2 under both normoxia and hypoxia. Conversely, mRNA levels of nNOS and iNOS do not appear to be similarly affected, indicating that eNOS may be the principal target of this system in H9c2. As a result of their control of NO production, LANCL1/2 expression levels directly affect cell glucose uptake and oxidation and the NADP/H content under normoxia, as well as mitochondrial proton gradient conservation after hypoxia/reoxygenation and ultimately cell vitality. Indeed, abrogation of all of these effects by L-NAME demonstrates the causal role of NO in the conservation of cell respiration and vitality downstream of the ABA/LANCL system.

#### 3.1.5. Signaling Downstream of LANCL1/2 Involves the AMPK/PGC-1α/Sirt1 Axis

In LANCL1/2-overexpressing H9c2, the AMPK/PGC-1α/Sirt1 axis is transcriptionally activated; conversely, a significant reduction of mRNA levels for these proteins is observed in LANCL1/2-silenced cardiomyocytes. PGC-1α is among the targets of activated AMPK. PGC-1α mediates the transcriptional outputs triggered by multiple metabolic sensors: it receives inputs from both AMPK and Sirt1, which can phosphorylate (AMPK) and deacetylate (Sirt1) PGC-1α, thus controlling its activity. Activated PGC-1α in turn stimulates mitochondrial biogenesis, increases cell respiration, and activates energy expenditure, particularly in the muscle. PGC-1α acts by co-activating other transcription factors, including glucocorticoid receptors (GRs), thyroid hormone receptor (TR), estrogen receptors (ERs), and estrogen-related receptors (ERRs). Moreover, PGC-1α is highly expressed in those tissues that show a high metabolic oxidative capacity, e.g., the heart, skeletal muscle, BAT, and brain, and its transcription is activated by cold, fasting, and exercise, all conditions that require an increased energy production. Downstream of Sirt1, the increased transcription of Nampt also occurs in LANCL1/2-overexpressing H9c2, indicating the functional as well as transcriptional activation of Sirt1 [34]. Overexpression of Nampt, or induction of Sirt1, protects rat heart from ischemia/reperfusion injury [38,39]. Conversely, in subjects with heart failure, a significant reduction of Sirt1 expression is observed in cardiomyocytes: indeed, the whole AMPK/Sirt1/Nampt axis appears to be downregulated in the aging/failing heart [40]. In view of these observations, upregulation of Nampt and of Sirt1 transcription in LANCL1/2-overexpressing cardiomyocytes is in line with a protective role of the LANCL proteins against hypoxia.

PGC-1α, the master regulator of mitochondrial function and biogenesis, also increases at the transcriptional level in LANCL1/2-overexpressing rodent skeletal myocytes and adipocytes, as well as cardiomyocytes, and is conversely reduced in LANCL1/2 double silenced cells. In L6 skeletal myocytes, LANCL1/2-overexpression stimulates mitochondrial respiration and the expression of skeletal muscle uncoupling proteins sarcolipin and UCP-3 [19]. In 3T3-derived adipocytes, treatment with ABA stimulates O_2_ consumption and induces the transcription of “browning genes” including UCP-1 [18]. Indeed, a dysfunctional AMPK/PGC-1α/Sirt1 signaling axis is considered to be responsible for reduced muscle energy expenditure, as occurs in aging and in metabolic disorders, such as T2D [41]. For this reason, pharmacological interventions capable of activating the AMPK/PGC-1α/Sirt1 axis are considered a promising strategy to protect muscle and heart function under conditions that reduce myocyte vitality (aging, hypoxia, diabetes) [42,43,44,45,46].

## 4. Neuroprotective Effects of the ABA/LANCL System

Neuroinflammatory processes induce neuronal damage and underlie the onset of neurodegenerative pathologies. Aging is the major factor associated with neurodegenerative diseases, but several conditions associated with chronic neuro-inflammation cause or aggravate neurodegeneration. In particular, diabetes mellitus is frequently associated with cognitive dysfunction, particularly in the elderly, and cognitive impairment and dementia are increasingly recognized as an important comorbidity of diabetes, which also affects the patient’s capacity to manage the disease. Indeed, recent guidelines recommend screening for cognitive impairment in diabetic patients [47].

On the contrary, physical exercise reduces neuroinflammation and improves memory, facilitating synaptic plasticity and neurogenesis [48,49]. It is a great challenge of modern societies to devise efficient strategies to prevent and treat neurodegeneration in the elderly. LANCL1 and LANCL2 are both highly expressed in the central nervous system (CNS) [19], where the glucose transporters comprise GLUT1 and GLUT4 [50,51], both targets of LANCL1/2-mediated increased expression. A role for LANCL1/2 in protection from neuroinflammation, particularly from its oxidation-mediated damage, could be hypothesized from the fact that both proteins bind to reduced glutathione (GSH) and have an SH3-binding domain [52]. These features have been proposed to allow these proteins to sense the cell redox state and be actors in response to its dysregulation.

### 4.1. LANCL1 Protects Neurons from Oxidative Stress In Vivo

Loss of LANCL1 leads to the accumulation of ROS, inflammation, mitochondrial dysfunction, and apoptotic neurodegeneration [53], whereas LANCL1 overexpression protects neurons against exogenous peroxide-induced apoptosis [54]. Indeed, LANCL1 overexpression protects motor neurons from apoptosis in mice with a genetic mutation of SOD, a model of amyotrophic lateral sclerosis, reportedly via the activation of Akt [54].

### 4.2. LANCL1 Protects Neurons against Death from Oxygen and Glucose Deprivation

Further results suggesting that LANCL1 plays a protective role in neurons were reported in an in vitro study on a model of neuronal cell death caused by oxygen- and glucose-deprivation (OGD). Overexpression of LANCL1 preserved cell viability, reduced lactate dehydrogenase (LDH) release, preserved mitochondrial function, and attenuated apoptosis after OGD in a neuronal cell line. These effects were mediated by the activation of an Akt/PGC-1α/Sirt3-dependent signaling pathway [55].

The high expression level of LANCL1/2 in the brain may be related to their stimulatory/protective effects on mitochondrial function.

### 4.3. ABA Improves Cognitive Impairment in Animal Models of Alzheimer’s Disease (AD)

Chronic treatment with ABA of transgenic mice homozygous for three mutant transgenes inducing AD (presenilin1; amyloid precursor protein Swe and tauP301L) significantly enhanced behavioral performance in comparison with untreated transgenic mice; in addition, in ABA-treated transgenic mice, microglia displayed a resting phenotype instead of the activated state observed before treatment, suggesting that ABA may protect cognitive function by preventing microglia activation [56]. Moreover, in five familiar AD (5 × FAD) mice, widely used as an AD animal model, ABA treatment improved cognitive and memory impairment and decreased Aβ deposition and neuroinflammation, reducing mRNA expression levels of pro-inflammatory cytokines, such as TNF-α, IL-1β, and IFN-γ. Interestingly, ABA treatment increased the protein level of LANCL2 in the cortex and hippocampus of ABA-treated 5 × FAD mice [57].

### 4.4. Transcriptional Activity of LANCL1/2

LANCL1 has been shown to bind to the SH3 domain of the signaling protein Eps8, an interaction necessary to allow nerve growth factor-induced neurite outgrowth in the model neuronal cells PC12 [52]. Eps8 is a target of phosphorylation by several growth factor receptors and is itself a transcriptional regulator with pro-proliferative actions, as witnessed by the tumor-inhibiting effect of peptides preventing its nuclear translocation [58]. Interestingly, nuclear translocation is also a feature of LANCL2, which detaches from the plasmamembrane upon ABA binding and migrates to the nucleus [15].

## 5. Conclusions and Future Perspectives

### 5.1. Conclusions

In diabetes mellitus, glycemia dysregulation and myocardial and neurological damage coexist; endogenous ABA deficiency could play a role in all of these pathological conditions, and it is tempting to speculate that exogenous ABA supplementation could therefore improve these conditions.

Indeed, the results summarized in this review provide a solid basis for the implementation of clinical studies aimed at evaluating the beneficial effect of nutraceutical ABA supplementation in diabetic patients, in conjunction with currently used drugs, which do not provide relief for endogenous ABA deficiency.

At the dose used in published clinical studies, ABA is devoid of side-effects and efficacious in reducing glycemia and lipidemia, both causes of vascular damage and tissue hypoxia. Low-cost vegetal extracts titrated in ABA can provide a dose of the hormone sufficient to exert its pharmacological effects. Oral ABA is promptly absorbed, devoid of hypoglycemic effects even at a dose hundreds of times higher than the effective dose (1 μg ABA/kg BW). Nutraceutical products titrated in ABA are already on the market, and clinical studies on diabetic patients could start immediately. Moreover, treatment with ABA prior to the development of overt diabetes is achievable in prediabetic subjects, and results from preliminary clinical studies [24,29] encourage us to pursue an ABA-based nutraceutical approach to reverse the prediabetic condition. Finally, several different vegetal sources already approved for human use as nutraceuticals can provide a dose of ABA in the range of therapeutic efficacy, i.e., approx. 1 µg/kg BW/day [17,30,59].

Physical activity, a physiological activator of AMPK, and metformin, a pharmacological activator of AMPK, are being prescribed to diabetic or prediabetic subjects to improve glucose tolerance, cardiac function, and neuroprotection. ABA is the endogenous hormone in charge of AMPK activation and activates the same signaling pathway as metformin, a currently used drug for the treatment of diabetes and prediabetes and also proposed for neuroprotection [60]. Instead of being an exogenous molecule, ABA is an endogenous physiological hormone, with pleiotropic functions, which can only be replicated by the molecule itself, if deficient. The multifaceted actions of the ABA/LANCL hormone/receptor system can indeed be hardly replicated by any single (pharmacological) compound known so far: ABA, via LANCL1 and LANCL2 and the downstream signaling axis involving AMPK and PGC-1α stimulates GLUT1/4-dependent glucose uptake in muscle and adipose tissue, it stimulates glucose oxidative metabolism, mitochondrial function (proton gradient), and adipocyte energy expenditure (thermogenesis), it increases muscle insulin sensitivity (insulin receptor expression) and white adipocyte browning, it improves physical performance and endurance (in mice), it protects cardiomyocytes from hypoxia-induced damage via increased NO production, and it stimulates mitochondrial biogenesis and respiration in cardiomyocytes (Figure 3). Further studies are needed to determine whether oral ABA has a cardioprotective effect in vivo, especially in diabetics. In fact, current strategies to reduce acute ischemia/reperfusion injury advocate the stimulation of glycolysis via AMPK activation, of mitochondrial glucose oxidation, of Nampt activity, and the elevation of the levels of Sirt1/Sirt3, all effects observed downstream of the ABA/LANCL signaling pathway in cardiomyoblasts [38,39].

ABA supplementation should improve and prolong the action of exogenous insulin, reducing the daily dose of insulin and the risk of hypoglycemia and ameliorating glycemic control in insulin-deficient patients, while ABA supplementation could contribute to the reduction of postprandial hyperglycemia, in conjunction with oral hypoglycemic drugs, by synergizing with the action of endogenous insulin, in insulin-resistant patients, via a different signaling pathway, eliciting increased muscle and adipocyte glucose uptake, without stimulating triglyceride accumulation in the adipose tissue. Pretreatment with ABA prior to the development of overt diabetes is achievable in prediabetic subjects, encouraging us to pursue an ABA-based nutraceutical approach to reverse the prediabetic condition.

### 5.2. Future Perspectives

A rapidly expanding field of exploration of mammalian ABA physiology is the role of ABA in the CNS. The fact that the brain has the highest ABA content among the various tissues [61] raises the possibility that ABA is produced and acts locally in the brain. Indeed, mounting evidence indicates that ABA administration improves neuroinflammation and cognitive impairment in rodents. The regular intake of the phytohormone can effectively prevent memory loss in a murine model of Alzheimer’s disease [62]. Chronic ABA administration also reduces brain proinflammatory cytokine expression and improves hippocampal neurogenesis in a rodent model of metabolic syndrome [63]. A further recent study showed that ABA elicits positive effects on harmaline-induced cognitive and motor disturbances in a rat model of essential tremor [64]. Interestingly, ABA administration improves cognitive performance in diabetic rats [65] and relieves anxiety and stress-induced cognitive disorders in mice [64,66]. The biochemical mechanism underlying the protective effect of LANCL1, particularly regarding a possible role for LANCL1 in stimulating glucose transport/mitochondrial respiration, remains to be investigated but should be attempted in light of these recent findings. Early treatment with ABA has been shown to ameliorate neuroinflammation and memory loss in a rodent model of Alzheimer’s [67]. A role for LANCL2 in the beneficial effect of ABA on murine Alzheimer’s has recently been described: ABA reduced amyloid β deposition, neuroinflammation, and memory impairment by the upregulation of LANCL2 [57]; the possibility that the activation of the ABA/LANCL system may reduce neuro-inflammation and the associated Aβ- and Tau-mediated pathology, preventing severe impairment of cognitive status, opens an unexplored scenario to contrast AD. Diabetes patients are at increased risk of developing cognitive impairment, dementia and Alzheimer’s disease [68,69,70]; thus, interventions targeting the ABA/LANCL system may afford a two-sided beneficial effect on the metabolic and also the cognitive dysfunctions caused by the disease.

Finally, an area of investigation that is beginning to be focused upon and that is likely to attract scientific interest in the future is the possible link between chronic neuroinflammation and SARS-CoV2 infection [71,72,73]. Cognitive impairment is among the clinical signs of “long-COVID”, i.e., clinical manifestations occurring months or even years after SARS-CoV2 infection. Along with other, still not precisely defined signs and symptoms, long-COVID is attributed to persistent tissue inflammation and possibly mitochondrial dysfunction [74]. Given the strikingly high number of SARS-CoV2-infected people world-wide (650 million and counting), if even a small percentage of this population should develop neurological signs of long-COVID-19, we might be in dire need for new treatments addressing this emergency in the future.

## Figures and Tables

**Figure 1 ijms-24-01199-f001:**
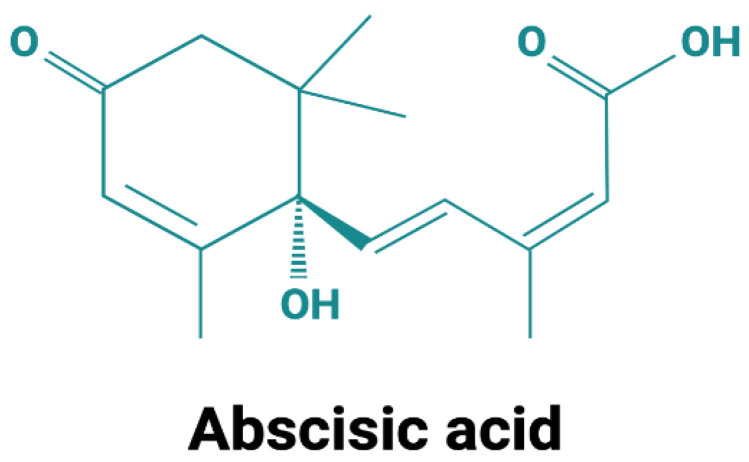
Structure of abscisic acid. Abscisic acid exists in two enantiomers: *S*-(+)-ABA, which is the predominant form in plants, and *R*-(−)-ABA. Most studies in mammals employed a racemic mixture. When individually tested on innate immune cells, both ABA enantiomers were similarly effective in the induction of an intracellular Ca^2+^ increase and in stimulating chemotaxis [1].

**Figure 2 ijms-24-01199-f002:**
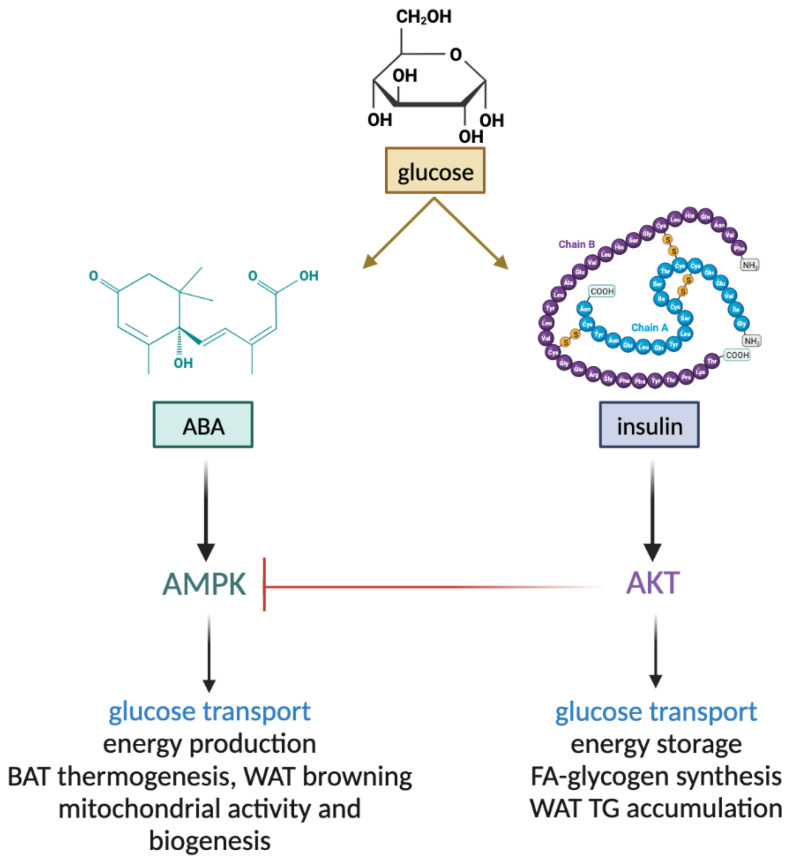
Non-overlapping functions of ABA and insulin in muscle and adipose tissue. ABA and insulin both stimulate glucose uptake by muscle and adipose tissue. Insulin, via Akt, stimulates the conversion of metabolic energy into storage forms, such as muscle glycogen and fatty acids and white adipocyte triglycerides. Activated Akt inhibits AMPK (blunted red line). Instead, ABA stimulates energy production via increased mitochondrial activity and biogenesis in muscle and adipose cells and thermogenesis. BAT, brown adipose tissue; WAT, white adipose tissue; FA, fatty acids; TG, triglycerides.

**Figure 3 ijms-24-01199-f003:**
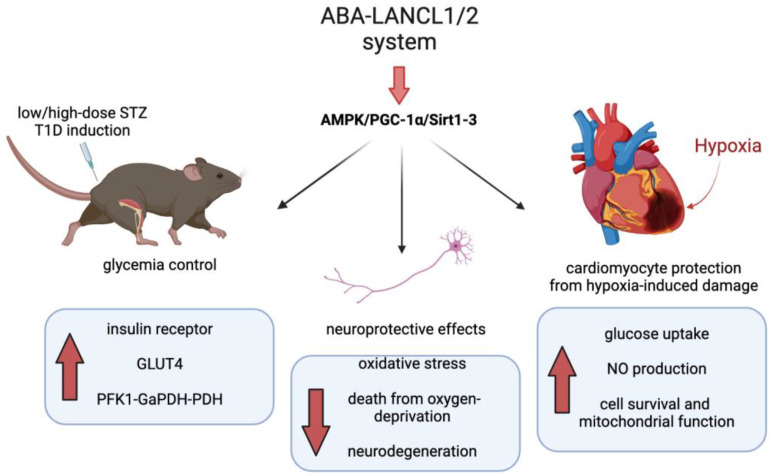
The ABA/LANCL system activates the AMPK/PGC-1α/Sirt1-3 axis. By activating the AMPK/PGC-1α/Sirt1-3 axis in myocytes and neuronal cells, the ABA/LANCL system impacts the fundamental physiological functions related to cell glucose uptake and metabolism, mitochondrial biogenesis and function, and protection from oxidative stress. These functions are all called upon to reduce the systemic effects of diabetes. Red arrows indicate increase (right and left panel) or decrease (central panel) of the indicated effects.

## Data Availability

Not applicable.

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
