# Peer review of "The ABA/LANCL Hormone/Receptor System in the Control of Glycemia, of Cardiomyocyte Energy Metabolism, and in Neuroprotection: A New Ally in the Treatment of Diabetes Mellitus?"

_ijms, 2023, doi:10.3390/ijms24021199_

Round 1

Reviewer 1 Report

The authors present a thorough review on the ABA-LANCL system and its effects in cell physiology as well as in several pathologies such as diabetes and neurodegenerative diseases.

Although the topic at hand is quite novel, he literature review is comprehensive and satisfactory, and the article itself is very well-written and easy to follow. The authors' focus on future perspectives contributes to making this review an important starting point for any investigator who would be interested in pursuing this avenue of research.

A few, very minor, points:

- The structure of Abscisic acid on page 2 appears pixelated, it should be redrawn.

- R- and S- , when used to denote the two enantiomers, should be in italics.

- In the caption of figure 1 the authors mention that S-ABA is the active isomer in plants: is it the same for animals?

- In the same caption, the sentence "...which differs from the S-isomer by a flip around the chiral carbon" is unnecessary.

A minor english spellcheck is also necessary.

All things considered, this article is more than acceptable for publication on IJMS.

Author Response

Reviewer # 1

Comments and Suggestions for Authors

The authors present a thorough review on the ABA-LANCL system and its effects in cell physiology as well as in several pathologies such as diabetes and neurodegenerative diseases.

Although the topic at hand is quite novel, the literature review is comprehensive and satisfactory, and the article itself is very well-written and easy to follow. The authors' focus on future perspectives contributes to making this review an important starting point for any investigator who would be interested in pursuing this avenue of research.

A few, very minor, points:

- The structure of Abscisic acid on page 2 appears pixelated, it should be redrawn.

The structure of ABA has been redrawn to improve image quality.

- R- and S-, when used to denote the two enantiomers, should be in italics.

Italics have been employed for R- and S-enantiomer identification.

- In the caption of figure 1 the authors mention that S-ABA is the active isomer in plants: is it the same for animals?

In mammals, most experiments were performed with a racemic mixture of both enantiomers. When they were individually tested (e.g. in Bruzzone S. et al. Proc Natl Acad Sci USA 2007;104(14):5759-64), a similar effect was observed. A sentence has been added to the legend of Figure 1 regarding this point.

- In the same caption, the sentence "...which differs from the S-isomer by a flip around the chiral carbon" is unnecessary.

The sentence has been removed.

A minor english spellcheck is also necessary.

We checked the whole manuscript again for errors.

All things considered, this article is more than acceptable for publication on IJMS.

Thank you!

Reviewer 2 Report

In their manuscript Spinelli et al. reviewed the ABA-LANCL hormone-receptor system in the control of glycemia, of cardiomyocyte energy metabolism and in neuro-protection. The Authors covered the topic quite extensively and described the properties of the hormone in mammalian organisms, with particular focus on adipose tissues, skeletal and cardiac muscle, and central nervous system in the context of metabolic dysregulation in obesity, diabetes and the associated co-morbidities. They also discussed the potential beneficial effects of ABA supplementation in animal models of brain diseases (e.g., Alzheimer disease). The manuscript is well written and offers a comprehensive overview of the ABA/LANCL receptor system.

I have a few minor suggestions to improve the manuscript:

1.     The Authors used the term “homology” with percentage values referring to the three LANCL proteins. Actually, quantifying homology is not appropriate. Proteins are either homologous or not, without percentage homology. This terminology is not appropriate since % identity is the observation, while homology is the conclusion. Thus, the correct term should be “% sequence identity” rather than “% sequence homology”. This is the case, for instance, at pg 2 line 64 and line 71.

2.     Pg 8 lines 332-335: the Authors stated “ABA appears to be mainly produced by β-cells, as plasma ABA is very low or undetectable in T1D subjects; thus, a severe reduction or the complete destruction of β-cells could affect the release of both hormones uniquely endowed with the ability to stimulate muscle glucose uptake.” On one hand, this interpretation is conceivable. However, one may argue that the lack of insulin in T1D may impair the production and release of ABA in tissues/organs other than pancreatic beta-cells. Therefore, the possibility that other tissues/organs produce ABA cannot be ruled out based solely on the observation that in T1D plasma ABA is low. Does administration of exogenous insulin restore, at least partially, the plasma levels of ABA in models of T1D (e.g., STZ-treated rodents)? Authors may want to consider this alternative in their discussion. At this regard, in a subsequent paragraph (pg 9 lines 380-396) the Authors mentioned studies reporting ABA release from cardiomyocytes under hypoxic conditions, thus indicating that different tissues can produce and release ABA.

3.     Pg 11 line 475: the Authors stated that GLUT1 and GLUT4 are expressed in the CNS. GLUT4 is mainly expressed in adipose tissues, heart, and skeletal muscles. It would be important to cite papers reporting the expression of GLUT4 in the brain or in specific brain areas.

4.     Perhaps an area worth discussing is the potential role of ABA/LANCL receptors in the outcome of SARS-CoV-2 infection. The reported beneficial effects on neuroinflammation and improvement of cognitive function in animal models treated with ABA, could suggest its possible use as a supplement to support the therapies for infected patients and to mitigate the long terms effects of post-COVID-19 infection (i.e., long COVID), in particular with those patients affected by impaired cognitive functions. The Authors could add a short paragraph if there is any literature available on this specific aspect.

Author Response

Reviewer # 2

Comments and Suggestions for Authors

In their manuscript Spinelli et al. reviewed the ABA-LANCL hormone-receptor system in the control of glycemia, of cardiomyocyte energy metabolism and in neuro-protection. The Authors covered the topic quite extensively and described the properties of the hormone in mammalian organisms, with particular focus on adipose tissues, skeletal and cardiac muscle, and central nervous system in the context of metabolic dysregulation in obesity, diabetes and the associated co-morbidities. They also discussed the potential beneficial effects of ABA supplementation in animal models of brain diseases (e.g., Alzheimer disease). The manuscript is well written and offers a comprehensive overview of the ABA/LANCL receptor system.

I have a few minor suggestions to improve the manuscript:

  1. The Authors used the term “homology” with percentage values referring to the three LANCL proteins. Actually, quantifying homology is not appropriate. Proteins are either homologous or not, without percentage homology. This terminology is not appropriate since % identity is the observation, while homology is the conclusion. Thus, the correct term should be “% sequence identity” rather than “% sequence homology”. This is the case, for instance, at pg 2 line 64 and line 71.

The sentence has been corrected.

  1. Pg 8 lines 332-335: the Authors stated “ABA appears to be mainly produced by β-cells, as plasma ABA is very low or undetectable in T1D subjects; thus, a severe reduction or the complete destruction of β-cells could affect the release of both hormones uniquely endowed with the ability to stimulate muscle glucose uptake.” On one hand, this interpretation is conceivable. However, one may argue that the lack of insulin in T1D may impair the production and release of ABA in tissues/organs other than pancreatic beta-cells. Therefore, the possibility that other tissues/organs produce ABA cannot be ruled out based solely on the observation that in T1D plasma ABA is low. Does administration of exogenous insulin restore, at least partially, the plasma levels of ABA in models of T1D (e.g., STZ-treated rodents)? Authors may want to consider this alternative in their discussion. At this regard, in a subsequent paragraph (pg 9 lines 380-396) the Authors mentioned studies reporting ABA release from cardiomyocytes under hypoxic conditions, thus indicating that different tissues can produce and release ABA.

The sentence has been rephrased, taking into consideration the correct suggestion by the reviewer. Indeed, several cell types appear to be able to release ABA under tissue-specific conditions of stimulation: e.g. hypoxia in cardiomyocytes, pro-inflammatory stimuli in innate immune cells. Regarding the experiment suggested by the reviewer, i.e. whether insulin administration to insulin-deficient mice restored plasma ABA levels, no experimental data are available regarding this issue, a possibility which is certainly worth exploring in the future. A sentence has been added to acknowledge this important suggestion (page 8, lines 348-349).

  1. Pg 11 line 475: the Authors stated that GLUT1 and GLUT4 are expressed in the CNS. GLUT4 is mainly expressed in adipose tissues, heart, and skeletal muscles. It would be important to cite papers reporting the expression of GLUT4 in the brain or in specific brain areas.

Two references (50 and 51) have been added to support this statement.

  1. Perhaps an area worth discussing is the potential role of ABA/LANCL receptors in the outcome of SARS-CoV-2 infection. The reported beneficial effects on neuroinflammation and improvement of cognitive function in animal models treated with ABA, could suggest its possible use as a supplement to support the therapies for infected patients and to mitigate the long terms effects of post-COVID-19 infection (i.e., long COVID), in particular with those patients affected by impaired cognitive functions. The Authors could add a short paragraph if there is any literature available on this specific aspect.

Although we could not find any literature, this is certainly a subject worth of investigation. A short paragraph has been added (page 13, line 632), with four new references (71-74).

I wish to express my personal gratitude to the Reviewer for the insightful comments, which not only improved the manuscript, but also suggest interesting areas of future investigation.